# Real-World Experience in the Clinical Use of Pirfenidone in Patients with Idiopathic Pulmonary Fibrosis in Taiwan: A Post-Marketing Surveillance Study

**DOI:** 10.3390/biomedicines12102348

**Published:** 2024-10-15

**Authors:** Ping-Hung Kuo, Chih-Yen Tu, Chia-Hung Chen, Kuo-Chin Kao, Jeng-Yuan Hsu, Meng-Chih Lin, Inn-Wen Chong, Chau-Chyun Sheu

**Affiliations:** 1Department of Internal Medicine, National Taiwan University Hospital, Taipei 100225, Taiwan; kph712@ntuh.gov.tw; 2Division of Pulmonary and Critical Care, Department of Internal Medicine, China Medical University Hospital, Taichung 404327, Taiwan; chesttu@gmail.com (C.-Y.T.); hsnu758@gmail.com (C.-H.C.); 3School of Medicine, College of Medicine, China Medical University, Taichung 406040, Taiwan; 4Department of Thoracic Medicine, Chang Gung Memorial Hospital, Taoyuan 33305, Taiwan; kck0502@cgmh.org.tw; 5Department of Respiratory Therapy, College of Medicine, Chang Gung University, Taoyuan 33302, Taiwan; 6Division of Chest Medicine, Department of Internal Medicine, Taichung Veterans General Hospital, Taichung 407219, Taiwan; tobyhsu312@gmail.com; 7Division of Pulmonary and Critical Care Medicine, Department of Internal Medicine, Kaohsiung Chang Gung Memorial Hospital, Chang Gung University College of Medicine, Kaohsiung 83301, Taiwan; mengchih@cgmh.org.tw; 8Division of Pulmonary and Critical Care Medicine, Department of Internal Medicine, Kaohsiung Medical University Hospital, Kaohsiung 80756, Taiwan; chong@kmu.edu.tw; 9Department of Internal Medicine, School of Medicine, College of Medicine, Kaohsiung Medical University, Kaohsiung 80708, Taiwan

**Keywords:** pirfenidone, idiopathic pulmonary fibrosis, Taiwan, safety, forced vital capacity, real-world registry

## Abstract

Background: Idiopathic pulmonary fibrosis (IPF) is a serious, progressive lung disease for which treatments are scarce. Pirfenidone has been approved for the treatment of IPF in Taiwan since 2016. This study aimed to gain a better insight into pirfenidone’s real-world safety and effectiveness in adult IPF patients in Taiwan. Methods: We conducted a prospective, multicenter, post-marketing surveillance study, and analyzed data from a small sample of 50 IPF patients treated with pirfenidone. Results: Most patients were men, with a mean age of 72.8 years (±10.3). They were in physiology stage I or II with a baseline mean forced vital capacity (FVC) of 2.236 L (73.8% of predicted value). After treatment with pirfenidone, the mean FVC decreased by 0.088 L at week 24 and 0.127 L at week 52. The mean 6 min walk test was 325.5 m at baseline, increased by 8.1 m at week 24, but then decreased by 23.0 m at week 52. These changes from baseline did not reach statistical significance. Pirfenidone prevented worsening of cough but did not stabilize dyspnea. During 52 weeks of treatment, the incidence of total adverse drug reactions was 62.0%, with decreased appetite (32.0%) and pruritis (10.0%) being the most common. The adverse events leading to treatment discontinuation were decreased appetite (8.0%), nausea (4.0%), and respiratory failure (4.0%). No safety concern was raised by the study. Treatment with pirfenidone stabilized both FVC and the subjective symptom of cough in most patients. Conclusions: This post-marketing surveillance study demonstrated that pirfenidone is an effective, safe, and well-tolerated treatment in patients with IPF in Taiwan.

## 1. Introduction

Idiopathic pulmonary fibrosis (IPF) is a progressive, irreversible, life-threatening interstitial lung disease (ILD). In Europe and North America, its incidence has risen over time and is estimated to be 3–9 cases per 100,000 persons per year [1]. The incidence and prevalence of IPF are lower in Taiwan than in Western countries. Analyses of Taiwan’s National Health Insurance (NHI) Research Database showed that the incidence remained stable after 2005, ranging from 0.7 to 1.3 cases per 100,000 persons per year, whereas the cumulative prevalence increased steadily from 3.1 to 6.4 cases per 100,000 persons per year during 2006–2011 [2]. The mean survival after diagnosis was 6.9 years. The average age of disease onset was 65 years, younger than that of Western countries, and men aged over 75 years had higher incidence compared with other age groups. The increasing prevalence and incidence might have been due to the increased utilization of diagnostic measures, especially high-resolution computed tomography (HRCT).

Without knowing the cause of IPF and its cures, treatment with antifibrotics, namely pirfenidone and nintedanib, is aimed at relieving symptoms, slowing its clinical progression, preventing acute exacerbation, and improving patients’ quality of life. Both drugs have shown promising results in extending survival as well. Pooled analyses of clinical trials showed that pirfenidone was associated with a significant reduction in mortality, respiratory-related hospitalization, and death after hospitalization [3,4], whereas nintedanib was associated with lower risks of on-treatment mortality and adverse events [5]. A Korean retrospective study using propensity score matching found that antifibrotic treatment significantly reduced the risks of mortality, all-cause hospitalization, respiratory-related hospitalization, acute exacerbation, and mortality after acute exacerbation [6]. Moreover, the real-life experience of a single center supported the efficacy and safety of pirfenidone in IPF patients, including improved lung function and progression-free survival [7]. Another real-life study also confirmed the efficacy and safety of pirfenidone in the case of the concomitant use of antithrombotic and/or anticoagulant drugs [8].

Since 2016, two antifibrotic drugs, pirfenidone and nintedanib, have been approved for the treatment of IPF in Taiwan. Ideally, clinicians should be well-informed of the use of these drugs for the treatment of IPF in terms of effectiveness, tolerability, and safety, particularly in a real-world setting. It has been reported that among IPF patients in Taiwan, pirfenidone and nintedanib stabilized lung function parameters over 2 years without increasing mortality or safety hazards, while preserving quality of life [9]. However, the majority of the treated patients in the analysis (83%, n = 73) were administered nintedanib, and only a few patients (5.7%, n = 5) were administered pirfenidone. The rest of them switched from pirfenidone to nintedanib due to various reasons. It is therefore necessary to gather additional local data on pirfenidone. A previous report focused mainly on pulmonary function measurements and did not provide a detailed description on the safety and tolerability of pirfenidone [10]. In this report, we acquired and analyzed data from pirfenidone’s only post-marketing surveillance (PMS) study in Taiwan, with the objective of gaining a better insight into its real-world safety and effectiveness in patients with IPF in the country.

## 2. Materials and Methods

### 2.1. Study Design

This was a prospective, observational, non-interventional, multi-center phase 4 PMS study to evaluate the safety and effectiveness in IPF patients treated with pirfenidone in a clinical setting. This study consisted of 32 weeks of recruitment period and 56 weeks of study period during which study data were recorded prospectively from February 2019 to January 2022. Recruitment was conducted in 6 study centers in Taiwan which had access to pirfenidone. The target number of patients to receive at least 1 dose of pirfenidone was 50. The end of the study was at the completion of treatment or the last follow-up of the last patient.

### 2.2. Inclusion and Exclusion Criteria

Included were male or female patients aged ≥20 years who were diagnosed with IPF using HRCT within 5 years before participating in the study. Only those patients who were naïve to pirfenidone, having forced vital capacity (FVC) values of ≥50% predicted, had undergone at least a 28-day wash-out period if treated with other medications for IPF, and who were covered under the NHI Insurance Pharmaceutical Benefits and Reimbursement Program for IPF were included in this study. Patients were excluded if they had other interstitial diseases, severe hepatic impairment, or end-stage renal disease, or were pregnant or breastfeeding (for female patients).

### 2.3. Pirfenidone Treatment

Pirfenidone (Pirespa^®^, manufactured and packaged by Shionogi & Co., Ltd., Osaka, Japan) was administered orally in a dose escalation manner according to the package insert—200 mg three times a day (TID) (600 mg/day) in the first 2 weeks, 400 mg TID (1200 mg/day) in the next 2 weeks, and 600 mg TID (1800 mg/day) afterwards until the end of week 52. Adverse drug reactions (ADRs) were managed by temporary dose reduction or treatment interruption until resolution of symptoms. For interruptions of less than 14 days, pirfenidone was resumed with the previous dose. For interruptions of more than 14 days, pirfenidone was reinitiated from the beginning dose, up to the full maintenance dose (600 mg TID). Patients were required to return to the study center at weeks 2 and 4 during the dose escalation period and then every 4 weeks until week 16. Thereafter, follow-ups were performed by alternating between telephone contacts and study center visits. Patients were required to return to the study center at week 52. Afterwards, there were an additional 4 weeks of safety follow-up done by telephone contacts.

### 2.4. Safety

Safety data were collected and analyzed for all patients. Adverse events (AEs), ADRs, serious AEs (SAEs), and AEs leading to dose modification/treatment discontinuation were summarized and tabulated by system organ class (SOC) and preferred term (PT).

### 2.5. Effectiveness

The effectiveness measures were changes in FVC, cough and dyspnea scores, oxygen saturation (SpO_2_), and 6 min walk test (6MWT). Categorical changes in FVC (improved [FVC increase ≥ 10%], stable [FVC change < 10%], or worsened [FVC decrease ≥ 10%]) and cough and dyspnea (improved [decreased score], stable [no change], or worsened [increased score]) were also summarized, as were the gender, age, and physiology (GAP) index and staging results.

The severity of cough was scored by a 4-point scale (1—no cough; 2—mild, intermittent cough; 3—moderate, irritating but not debilitating cough; 4—heavy, debilitating cough characterized by shortness of breath and exhaustion) as previously described [11]. The scoring of dyspnea was based on the Shortness of Breath Questionnaire from the University of California San Diego (UCSD SOBQ).

### 2.6. Statistical Analysis

Due to the study nature, the target sample size of 50 patients was not derived from statistical estimating method. All data of the study outcomes were analyzed descriptively. Continuous variables were summarized by number of observations, mean, standard deviation (SD), median, quartiles, minimum, and maximum values. The one-sample *t*-test or the Wilcoxon signed-rank test was used to compare changes from baseline values. Categorical variables were summarized by frequency and percentage of patients and by category. All data were documented using summary tables and patient data listings.

The following populations were applied in the analysis:Safety population (n = 50): all patients who received at least 1 dose of pirfenidone.Effectiveness-evaluable population (EEP) (n = 34): all patients who received at least 6 months of pirfenidone treatment and had at least one post-baseline effectiveness datum on either FVC, cough, dyspnea, or 6MWT.

## 3. Results

### 3.1. Baseline Characteristics and Use of Pirfenidone

Demographic and baseline characteristics of all patients are summarized in Table 1. The mean (SD) age was 72.8 (10.3) years. The majority of the patients were male (84.0%). All patients were Asian. Most patients were at GAP stage I (38.8%) or stage II (49.0%) (Figure 1), and were diagnosed with IPF within a year prior to the study (76.0%). The baseline mean (SD) FVC was 73.8% (14.7) predicted or 2.236 L (0.6). The majority of patients had ALT (92.0%) and AST (90.0%) levels ≤ 1 × upper limit of normal (ULN) at baseline.

All patients reported medical histories which mostly included medical conditions that were still ongoing at the time of participation. A total of 49 (98.0%) patients reported using medications prior to the study. They included cysteine derivative (48.0%), acetylcysteine (48.0%), theophylline (22.0%), and benzonatate (20.0%). A total of 42 (84.0%) patients reported using concomitant medications during the study. They included paracetamol (30.0%), famotidine (22.0%), and cysteine derivative (20.0%).

The majority of the patients received a daily dose of pirfenidone in the range of 601 mg/day to ≤1200 mg/day (28.0%) or 1201 to ≤1800 mg/day (54.0%), and none received it above the maximum daily dose (1800 mg/day) (Figure 2a). Of all patients, 38.0% received pirfenidone for 181 days to ≤360 days and 30.0% received it for more than 361 days (Figure 2b). The median duration of treatment of all patients was 336.0 days. Pirfenidone was the only antifibrotic drug used during the study.

### 3.2. Safety

A total of 42 (84.0%) patients reported a total of 128 AEs during the study, with most of them being mild (94 AEs reported in 25 [50.0%] patients) or moderate (18 AEs reported in 7 [14.0%] patients), as shown in Table 2.

Of all patients, 31 (62.0%) reported 56 AEs that were assessed by the investigator as related to the study drug. The most frequently reported ADRs were decreased appetite (32.0%) and pruritus (10.0%). Only one case (2%) of photosensitivity was reported. A total of 20 (40.0%) patients reported 28 AEs that led to dose modification or interruption. The frequently reported AEs leading to dose modification or interruption by SOC were metabolism and nutrition disorders (eight AEs reported in eight [16.0%] patients), skin and subcutaneous tissue disorders (six AEs reported in four [8.0%] patients), and gastrointestinal disorders (four AEs reported in four [8.0%] patients). The frequently reported AEs leading to dose modification or interruption by PT were decreased appetite (eight AEs reported in eight [8.0%] patients) and pruritus (four AEs reported in four [8.0%] patients). Only one patient (2%) had hepatic function abnormality leading to dose modification or interruption.

A total of 19 AEs leading to treatment discontinuation were reported in 14 (28.0%) patients. The frequently reported AEs leading to treatment discontinuation by SOC were metabolism and nutrition disorders (five AEs reported in five [10.0%] patients) and gastrointestinal disorders (four AEs reported in three [6.0%] patients). The frequently reported AEs leading to treatment discontinuation by PT were decreased appetite (four AEs reported in four [8.0%] patients), nausea (two AEs reported in two [4.0%] patients), and respiratory failure (two AEs reported in two [4.0%] patients). Eight (16.0%) patients reported 11 SAEs during the study, none of which were assessed as related to the study drug. Ten (20.0%) patients reported 16 AEs ≥ grade 3 during the study, and none were assessed as related to the study drug. Five patients (10.0%) died during the study. Their deaths were assessed as either unlikely to be related or not related to the study drug. During the study, three (6.0%) patients reported at least one episode of idiopathic acute exacerbation for which no underlying trigger was identified.

The mean values of all measured hematology and biochemistry parameters generally remained stable from baseline throughout the study.

### 3.3. Effectiveness

#### 3.3.1. FVC

The mean changes from baseline to week 24 and week 52 in the FVC values and categorial change from baseline to week 52 are summarized in Table 3. The mean (SD) baseline FVC was 75.98% (15.2) predicted or 2.23 L (±0.56). At week 24, mean (SD) FVC decreased by 0.53% (14.08) predicted or by 0.088 L (0.34); at week 52, mean (SD) FVC decreased by 1.01% (13.70) predicted or by 0.127 L (0.30) (Figure 3). In terms of categorical changes, at week 52, FVC improved in four (15.4%), remained stable in fifteen (57.7%), and worsened in seven (26.9%) patients.

#### 3.3.2. Cough and Dyspnea

The mean (SD) cough score at baseline was 2.1 (0.3) and decreased slightly throughout the study. At week 52, the mean (SD) cough score decreased only by 0.3 (0.6). In terms of categorical changes, at week 52, the cough score improved in eight (28.6%) and remained stable in twenty (71.4%) patients (Figure 4). None showed worsening in the score. On the other hand, at a mean (SD) baseline value of 30.0 (31.34), the dyspnea score decreased only for 4 weeks and increased afterwards. At week 52, the mean (SD) dyspnea score increased by 6.2 (19.31), which did not reach statistical significance (*p* = 0.116). Categorically, at week 52, the dyspnea score improved in eight (34.8%), remained stable in three (13.0%), and worsened in twelve (52.2%) patients (Figure 4).

#### 3.3.3. SpO_2_, 6MWT, and GAP Staging

There was minimal change of SpO_2_ throughout the study. At week 52, the mean (SD) SpO_2_ decreased only by 0.1% (2.5) (*p* = 0.883). The mean (SD) 6MWT was 325.5 m (133.6) at baseline, increased by 8.1 m (98.8) at week 24 (*p* = 0.807), but then decreased by 23.0 m (85.7) at week 52 (*p* = 0.203). At baseline, all except one patient were at GAP stage I (n = 15, 45.5%) or stage II (n = 17, 51.5%). At week 24 and week 52, three (9.1%) and four (15.4%) patients were at stage III, respectively. At week 52, the numbers of patients at Stage I and II were both 11 (42.3%) (Figure 5).

## 4. Discussion

Following the marketing approval of pirfenidone in May 2016 as the first pirfenidone treatment for IPF patients in Taiwan, this is the second published study providing real-world insights into the use of pirfenidone in IPF patients in the country. Pirfenidone has been included in Taiwan’s NHI Reimbursement Program since July 2017. Our analysis again confirmed the safety and effectiveness of this approved IPF treatment in this Asian population.

Compared to the previous report of pirfenidone use in Taiwanese IPF patients [10], we provided more detailed descriptions on its safety and tolerability. Generally, the use of the pirfenidone did not raise major safety concerns in the study. Most patients (84.0%) reported a total of 128 AEs during the study, of which 56 (43.8%) were drug-related. Most AEs were mild in severity. Eight (16.0%) patients reported 11 SAEs, of which none were related to the study drug, nor were the five deaths during the study. ADR rate was 62.0% in the study, which was similar to those reported in another Taiwanese cohort (n = 50, 52.0%) [10], a Korean cohort (n = 219, 69.0%) [12], and a larger Japanese PMS study (n = 1371, 64.6%) [13]. However, the AE-related discontinuation rate of 28% was higher than that of the Taiwanese cohort (4%) but similar to those of both the Korean and Japanese cohorts (22.8–24.3%). Given the similarity of disease severity between this cohort and theirs (FVC 73.8% vs. 65.0–66.4% predicted), it appeared that patients with IPF in Taiwan, compared to their neighbor counterparts, were not more likely to experience ADRs or discontinue pirfenidone during treatment. In this study, the most frequent ADRs were decreased appetite (32.0%) and pruritus (10.0%), which were consistent with other cohorts. Notably, only one case (2.0%) of photosensitivity was reported. These (incidences of pruritus and photosensitivity), combined with the incidence of rash (8.0%), yield an incidence of 20%, which is similar to the “ADR-dermatologic” incidence of 28% reported in the other Taiwanese cohort [10].

During the study period, Taiwan’s NHI reimbursed antifibrotic therapy for IPF patients with baseline FVC values of 80% or below. As part of the inclusion criteria, all patients included in this study represent typical real-world patients with IPF undergoing antifibrotic therapy in Taiwan. As clinical trials have shown that a ≥10% decline in predicted FVC is associated with mortality, change in FVC is frequently selected as a primary endpoint in phase 3 trials of antifibrotic therapies [14,15]. In patients in this study, the mean FVC change from baseline was −0.127 L or −1.01% predicted after 12 months of treatment. The mean reduction from baseline in FVC was slightly greater than that observed in the SP2 trial (−0.03 L at 9 months) [16] and SP3 trial (−0.08 to −0.09 L at week 52) [17] studies. Moreover, categorical assessment of FVC changes showed that 73.1% of patients in this study were assessed as stable or improved. This is similar to those reported in the Japanese PMS study (73.7%) [13] as well as the Japanese phase III trial (72.4%) [17]. It is commendable that the real-world treatment effectiveness can approximate that achieved in a well-controlled clinical trial. During 12 months of treatment, acute exacerbations were reported in only three (6.0%) patients. Two of the patients completed the study without any modifications to the dosage due to the acute exacerbation events. The remaining one patient had the dosage reduced and eventually discontinued treatment due not to acute exacerbation but decreased appetite.

Symptom management is a crucial issue for the daily living of patients with IPF. Our results showed that pirfenidone prevented worsening of cough in all patients. This aspect of symptom control is important because cough has been found to predict disease progression of IPF independent of disease severity [18]. Using a validated cough outcome tool (Leicester Cough Monitor and Leicester Cough Questionnaire), it has been found that pirfenidone treatment in IPF patients led to a significant reduction in objective cough frequency and clinically significant improvement in cough-related quality of life [19]. On the other hand, in contrast to the larger-scale Japanese PMS study, this small-scale PMS study did not show stabilization of dyspnea symptoms with long-term treatment. More than 50% of patients experienced worsening of dyspnea scores after 12 months. The possibility of this contrasting finding being caused by the small sample cannot be ruled out (see below). In fact, it is crucial that patients with IPF are managed in a multidisciplinary fashion, with antifibrotics forming one of the components [20]. Other components include smoking cessation, regular exercise/pulmonary rehabilitation, supplemental oxygen, management of comorbidities, nutrition, and so on. A meta-analysis showed that exercise training combined with breathing exercise produced a significant reduction in dyspnea in IPF patients [21]. Whether such comprehensive management strategies result in better dyspnea control in Taiwanese IPF patients remains to be elucidated.

Another aspect of our analysis worth mentioning is the 6MWT. Its mean change from baseline was 8.13 m at week 24 but −22.96 m at week 52, although its actual mean values at week 24 and week 52 were 323.25 m and 348.68 m, respectively, compared to 325.48 m at baseline. Actually, the mean 6MWT at week 52 showed no significant changes from baseline. The contrasting observation between the mean values and mean changes from baseline might be attributed to the small sample size and the huge variability in the results of 6MWT assessment. In fact, consistent with the mean 6MWT at week 52, the mean FVC at week 52 also remained stable versus baseline (77.68% vs. 75.98%). The mean SpO_2_ values and GAP stages over time were generally stable.

It should be emphasized that this study was limited by the small sample size and huge variability in the results, particularly in dyspnea scores and 6MWT data. Due to the nature of a PMS study, only a single arm of treatment with pirfenidone was included. No formal calculation of the sample size was performed, and the target number of patients was based on the accessible patient pool at the study sites.

## 5. Conclusions

This study provides real-world data on the use of pirfenidone in patients with IPF in Taiwan. Over 12 months of treatment, pirfenidone prevented worsening of lung function and cough, with tolerable side effects. No major safety concern was raised by the study. The most common adverse effects were decreased appetite and dermatologic symptoms. Future studies with larger patient datasets are warranted to elucidate the comprehensive roles of this antifibrotic therapy in patients with IPF in this part of Asia.

## Figures and Tables

**Figure 1 biomedicines-12-02348-f001:**
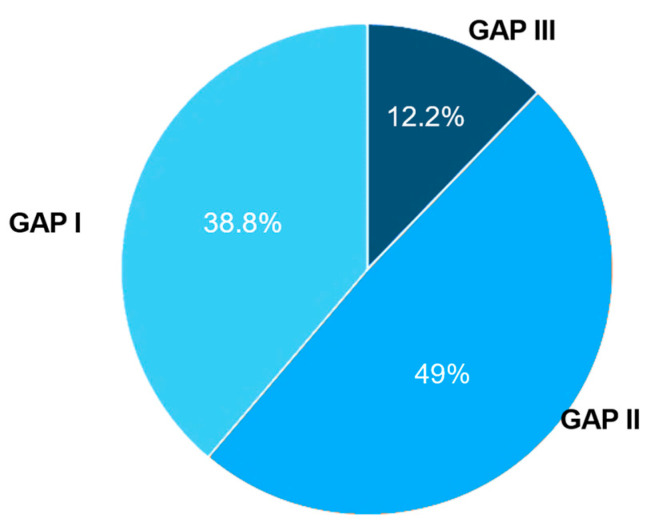
GAP staging of all patients at baseline.

**Figure 2 biomedicines-12-02348-f002:**
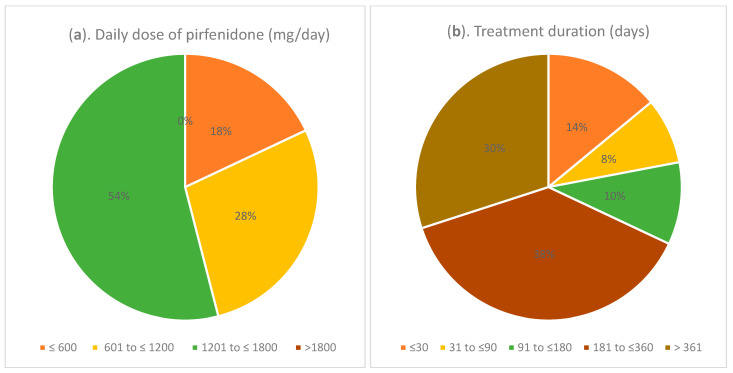
(**a**) Daily dose of pirfenidone (mg/day), (**b**) treatment duration (days).

**Figure 3 biomedicines-12-02348-f003:**
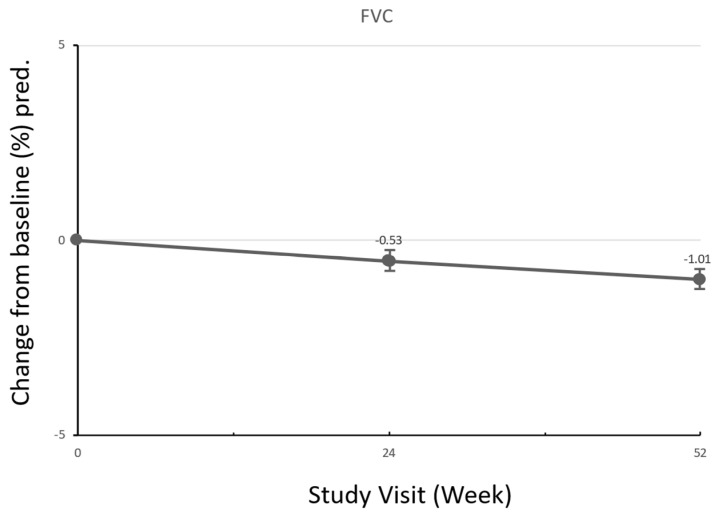
FVC change from baseline (% pred). FVC, forced vital capacity.

**Figure 4 biomedicines-12-02348-f004:**
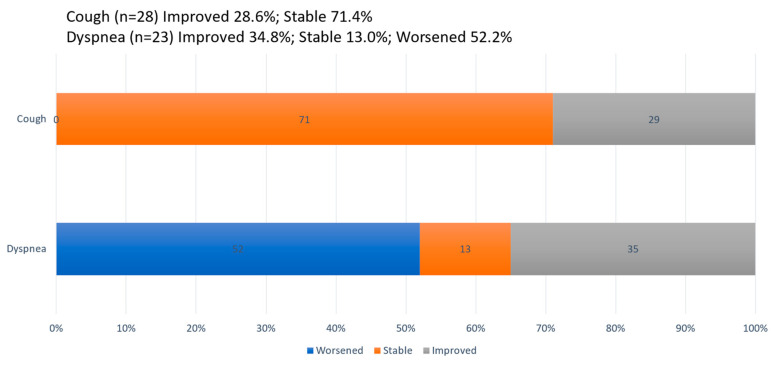
Categorical changes in subjective symptoms from baseline to week 52 were defined as “Improved” (grey, decrease score), “Stable” (orange, no change), or “Worsened” (blue, increased score).

**Figure 5 biomedicines-12-02348-f005:**
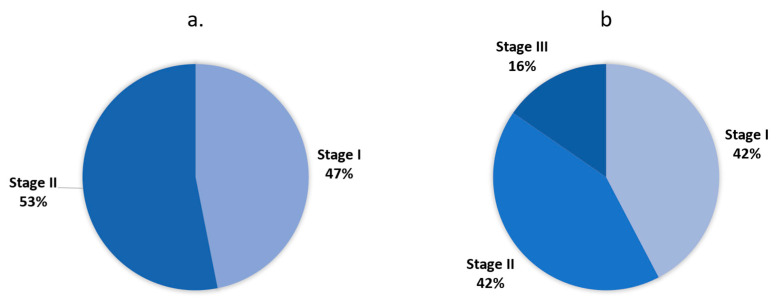
GAP staging of the effectiveness-evaluable population at baseline (**a**) and week 52 (**b**).

**Table 1 biomedicines-12-02348-t001:** Demographic and baseline characteristics and clinical use of pirfenidone of all patients (safety population, n = 50).

Parameter	Category	Statistics
Age (years)	mean (±SD)	72.8 (±10.26)
	median [25th, 75th percentile]	72.0 [65.0, 81.0]
Gender	Male	42 (84.0)
Race	Asian	50 (100.0)
BMI (kg/m^2^)	mean (±SD)	24.79 (±3.82)
	median [25th, 75th percentile]	24.38 [21.89, 27.78]
Time since first diagnosis (years)	<1	38 (76.0)
	1 to <3	6 (12.0)
	≥3	1 (2)
	Unknown	5 (10.0)
Previous drug (treatment for IPF)	Steroid	21 (42.0)
	Immunosuppressant	1 (2.0)
	Cysteine derivative	24 (48.0)
Most frequent daily dose administered (mg/day)	≤600	9 (18.0)
	601 to ≤1200	14 (28.0)
	1201 to ≤1800	27 (54.0)
	>1800	0
Treatment duration (days)	mean (±SD)	254.4 (138.33)
	median [25th, 75th percentile]	336.0 (114.0, 364.0)
FVC (L)	mean (±SD)	2.24 (±0.60)
	median [25th, 75th percentile]	2.10 [1.84, 2.59]
FVC (% predicted)	mean (±SD)	73.80 (±14.69)
	median [25th, 75th percentile]	75.05 [64.0, 79.1]
ALT (U/L)	≤1 × ULN	46 (92.0)
	>1 to ≤3 × ULN	3 (6.0)
	>3 to ≤5 × ULN	1 (2.0)
AST (U/L)	≤1 × ULN	45 (90.0)
	>1 to ≤3 × ULN	5 (10.0)

Data are presented as No. (%), unless otherwise stated. ALT, alanine aminotransferase; AST, aspartate aminotransferase; BMI, body mass index; FVC, forced vital capacity; SD, standard deviation; ULN, upper limit of normal.

**Table 2 biomedicines-12-02348-t002:** Most frequently reported adverse events (>5% of total patients) by system organ class and preferred term.

System Organ Class Preferred Term	Total Patients (n = 50)	
	Number of Events	n (%) Patients
Patients with any AEs	128	42 (84.0)
Metabolism and nutrition disorders	21	21 (42.0)
Decreased appetite	19	19 (38.0)
Gastrointestinal disorders	19	15 (30.0)
Abdominal pain upper	3	3 (6.0)
Diarrhea	4	4 (8.0)
Nausea	3	3 (6.0)
Skin and subcutaneous tissue disorders	19	12 (24.0)
Pruritus	9	8 (16.0)
Rash	6	4 (8.0)
Respiratory, thoracic, and mediastinal disorders	19	11 (22.0)
Dyspnea	3	3 (6.0)
Nervous system disorders	8	7 (14.0)
Dizziness	5	4 (8.0)
Injury, poisoning, and procedural complications	8	7 (14.0)
General disorders and administration site conditions	7	6 (12.0)
Investigations	7	5 (10.0)
Infections and infestations	5	5 (10.0)
Musculoskeletal and connective tissue disorders	5	4 (8.0)
Psychiatric disorders	3	3 (6.0)

AE, adverse event.

**Table 3 biomedicines-12-02348-t003:** Mean changes over time and categorical changes from baseline in forced vital capacity in the effectiveness-evaluable population.

	Baseline	n	Change from Baseline to Week 24	n	Change from Baseline to Week 52	n
FVC (% predicted)	75.98 (±15.19)	34	−0.53 (±14.08) *	33	−1.01 (±13.70) #	26
	77.0 [64.30, 87.00]		−0.60 [−8.00, +8.00]		−0.60 [−7.00, +9.00]	
FVC (L)	2.16 (±0.59)	34	−0.09 (±0.34)	33	−0.13 (±0.30)	26
	2.10 [+1.85, +2.59]		−0.08 [−0.23, +0.12]		−0.11 [−0.34, +0.03]	

Data are presented as mean (±SD); SD, standard deviation; median [25th percentile, 75th percentile]. FVC, forced vital capacity; max, maximum; min, minimum. The categorical changes in FVC were defined as “improved” (FVC increase ≥ 10%), “stable” (FVC change < 10%), or “worsened” (FVC decrease ≥ 10%). * *p* = 0.829, # *p* = 0.710.

## Data Availability

The data presented in this study are available on request from the corresponding author.

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
