# Peer review of "Real-World Experience in the Clinical Use of Pirfenidone in Patients with Idiopathic Pulmonary Fibrosis in Taiwan: A Post-Marketing Surveillance Study"

_biomedicines, 2024, doi:10.3390/biomedicines12102348_

Round 1
Reviewer 1 Report
Comments and Suggestions for Authors
In the Article titled: Real-World Experience in the Clinical Use of Pirfenidone in Patients with Idiopathic Pulmonary Fibrosis in Taiwan: A 3 Post-Marketing Surveillance Study.
Only descriptive analysis was performed, if the authors aim to analyze the safety and efficacy, the parameters evaluated should be compared at baseline and at the end of the study, perhaps a student t test could be applied, but at the moment no significance could be determined.
Why not remove the parameter of pregnancy test in Table 1 if none of the patients would apply since it is an exclusion criterion.
The authors aim to measure “real-world” safety and effectiveness, yet the study lacks various parameters. An efficacy study will determine whether the intervention produces the desired result, usually under strict ideal circumstances, preferably compared against a placebo group. While effectiveness studies measure the desired effect in a “real-world” setting, and should be compared against the standard intervention, in this case the standard pharmacological treatment. In this study there is no control or group to compare against, and if it’s in a “real-world” setting it would be an effectiveness study not of efficacy. There are existing standard treatments for IPF that the authors could have used as a control group.
The authors claim that this study confirms the safety and efficacy of perfinidone as a treatment for IPF, but no statistical significance was observed or even an analysis performed, furthermore, one of the primary predictive tools for IPF is the GAP score, and at the end of the study there was a progression in the staging of the patients at the end of the study. FVC is also a parameter used in the study, and 57.7% of the patients remained stable in this parameter while 26.9% had a decrease, indicating a worse prognosis and higher 1-year mortality, so I fail to see how the authors could claim the efficacy of the treatment.
The authors also claim to se a significant improvement in the symptoms, while no statistical test was performed, and even though some patients report an improvement in coughing, 52.2% report a worsening in dyspnea. The authors also claim a modest improvement in the 6MWT from baseline to 52 weeks, though not statistically proven.
The conclusions the authors reach can’t be proven with the data shown, no preserved lung function was demonstrated and improvement in coughing is not enough to justify the use of the drug.

Author Response
We greatly appreciate the valuable feedback provided by the reviewers. In response to the statistical issues raised, we are currently applying different statistical methods to reanalyze our data. However, we would require a few more weeks to incorporate the necessary revisions into the manuscript since many of our authors are currently attending the 2024 ERS Congress in Vienna. May we kindly request an extension for the submission of the revised version?
Point-by-point response to Comments and Suggestions for Authors:
Comments 1: Why not remove the parameter of pregnancy test in Table 1 if none of the patients would apply since it is an exclusion criterion.
Response 1: Thank you for pointing this out. We agree with this comment and will remove the pregnancy test parameter from Table 1 in the revised manuscript.
Comments 2: The authors aim to measure “real-world” safety and effectiveness, yet the study lacks various parameters. An efficacy study will determine whether the intervention produces the desired result, usually under strict ideal circumstances, preferably compared against a placebo group. While effectiveness studies measure the desired effect in a “real-world” setting, and should be compared against the standard intervention, in this case the standard pharmacological treatment. In this study there is no control or group to compare against, and if it’s in a “real-world” setting it would be an effectiveness study not of efficacy. There are existing standard treatments for IPF that the authors could have used as a control group.
Response 2: We agree with this comment. Since we currently do not have a control group, we will revise the manuscript accordingly by changing the term "efficacy" to "effectiveness." We will also include a comparison of our result with previous studies in the revised discussion section.
Comments 3: The authors claim that this study confirms the safety and efficacy of perfinidone as a treatment for IPF, but no statistical significance was observed or even an analysis performed, furthermore, one of the primary predictive tools for IPF is the GAP score, and at the end of the study there was a progression in the staging of the patients at the end of the study. FVC is also a parameter used in the study, and 57.7% of the patients remained stable in this parameter while 26.9% had a decrease, indicating a worse prognosis and higher 1-year mortality, so I fail to see how the authors could claim the efficacy of the treatment.
Response 3: We would like to clarify that for patients with IPF, no current drug therapy can completely stop disease progression or the decline in FVC. The two existing antifibrotic drugs (pirfenidone and nintedanib) are known to slow the rate of FVC decline. Although our data shows a mean FVC change from baseline of -0.127L (-1.01%), with 26.9% of patients experiencing a decrease in FVC after 12 months of treatment, this still suggests the effectiveness of pirfenidone, as the decline is less than what has been reported in patients who did not receive antifibrotic therapy in previous studies. In our revised manuscript, we will expand the discussion and include additional references from the literature to support this claim.
We appreciate your understanding and look forward to hearing from you regarding the extension request. Thank you again for your guidance and support.
Reviewer 2 Report
Comments and Suggestions for Authors
The topic is interesting and the paper is quite well written. The article covers an interesting and current topic. Nevertheless, in my opinion, some parts need to be improved, I have some comments:
1) Abstract. Most patients were men, with a mean age of 72.8 years (±10.3). They were in physiology stage I or 25 II with a baseline mean forced vital capacity (FVC) of 2.236 L (73.8% of predicted value). After treat- 26 ment with pirfenidone, the mean FVC decreased by 0.088 L at week 24 and 0.127 L at week 52. The 27 mean 6-minute walk test was 325.5 m at baseline, increased by 8.1 m at week 24, but then decreased 28 by 23.0 m at week 52. During 52 weeks of treatment, the incidence of total adverse drug reactions 29 was 62.0% with decreased appetite (32.0%) and pruritis (10.0%) being the most common. The ad- 30 verse events leading to treatment discontinuation were decreased appetite (8.0%), nausea (4.0%) and 31 respiratory failure (4.0%). No safety concern was raised by the study. Please, underline the most important statistically significant values to support the data.
2) Treatment with pirfenidone 32 stabilized both FVC and the subjective symptom of cough in most patients. We conclude that 33 pirfenidone is safe, well-tolerated treatment for preserving lung function and preventing worsening 34 of cough in IPF patients in Taiwan. Abstract might be beneficial to include a sentence in the abstract that briefly summarizes the key findings of the study. This can provide readers with a quick overview of the research.
3) The mean survival after diagnosis was 6.9 years. The average 47 age of disease onset was 65 years, younger than that of Western countries, and men aged 48 over 75 years had higher incidence compared with other age groups. The increasing prev- 49 alence and incidence might have been due to the increased utilization of diagnostic 50 measures especially high-resolution computed tomography (HRCT). 51 Without knowing the cause of IPF and its cures, treatment with antifibrotics, namely 52 pirfenidone and nintedanib is aimed at relieving symptoms, slowing its clinical progres- 53 sion, preventing acute exacerbation, and improving patients’ quality of life. Both drugs 54 have shown promising results in extending survival as well. Pooled analyses of clinical 55 trials showed that pirfenidone was associated with a significant reduction in mortality, 56 respiratory-related hospitalization, and death after hospitalization [3,4], whereas 57 nintedanib was associated with lower risks of on-treatment mortality and adverse events 58 [5]. A Korean retrospective study using propensity score matching found that antifibrotic 59 treatment significantly reduced the risks of mortality, all-cause hospitalization, respira- 60 tory-related hospitalization, acute exacerbation, and mortality after acute exacerbation [6]. 61 Since 2016, two antifibrotic drugs, pirfenidone and nintedanib have been approved 62 for the treatment of IPF in Taiwan. Ideally, clinicians have to be well-informed of the use 63 of these drugs for the treatment of IPF in terms of efficacy, tolerability, and safety partic- 64 ularly in a real-world setting. It has been reported that among IPF patients in Taiwan, 65 pirfenidone and nintedanib stabilized lung function parameters over 2 years without in- 66 creasing mortality or safety hazards, while preserving quality of life [7]. Please, improve this paragraph. Furthermore, I suggest to add some references to support the sentences, such as:
- Real-life experiences in a single center: efficacy of pirfenidone in idiopathic pulmonary fibrosis and fibrotic idiopathic non-specific interstitial pneumonia patients. Ther Adv Respir Dis. 2020 Jan-Dec;14:1753466620963015. doi: 10.1177/1753466620963015.
- Pirfenidone in Idiopathic Pulmonary Fibrosis: Real-World Observation on Efficacy and Safety, Focus on Patients Undergoing Antithrombotic and Anticoagulant. Pharmaceuticals (Basel). 2024 Jul 11;17(7):930. doi: 10.3390/ph17070930.
4) In this report we acquired and analyzed data from 73 pirfenidone's post-marketing surveillance (PMS) study, with the objective of gaining a 74 better insight into its real-world safety and efficacy in patients with IPF in Taiwan. Please, underline the novelty of the study.
5) 4. Discussion 236 Following the marketing approval of pirfenidone in May 2016 as the first pirfenidone 237 treatment for IPF patients in Taiwan, this is the second published study providing real- 238 world insights into the use of pirfenidone in IPF patients in the country. Pirfenidone has 239 been included in the Taiwan’s NHI Reimbursement Program since July 2017. The discussion section needs to be improved. It is necessary to underline the results obtained and compare them with previous or similar published literature.
Comments on the Quality of English LanguageMinor changes of English language are required.
Author Response
COMMENT #1,2
“Abstract might be beneficial to include a sentence in the abstract…”
RESPONSE #1,2:
The abstract is amended accordingly. Of note, effectiveness is now used instead of efficacy because of the lack of control. The lack of statistical significance is now stated on Lines 30 and 31.
COMMENT #3:
“Please, improve this paragraph. Furthermore, I suggest…”
RESPONSE #3: Two references (#7 and 8) are added to strengthen the introduction. Please see Lines 63-67.
COMMENT #4:
“Please, underline the novelty of the study.”
RESPONSE #4:
Now the “only PMS” is highlighted (Line 79).
COMMENT #5:
“It is necessary to underline the results obtained and compare them with…”
RESPONSE #5:
Relevant content can be located on Lines 255-288.
Reviewer 3 Report
Comments and Suggestions for Authors
The manuscript analyzes the real-world use of pirfenidone in treating idiopathic pulmonary fibrosis (IPF) in Taiwan. It details a post-marketing surveillance study with 50 adult patients to assess the drug's safety and efficacy. The study revealed that pirfenidone was generally well-tolerated, with the most common side effects being decreased appetite and pruritus. Despite mild declines in lung function, as measured by forced vital capacity (FVC), the drug helped stabilize symptoms in many patients. The article concludes that pirfenidone is a safe, effective option for managing IPF in Taiwan. The conclusions of the current manuscript are limited by the small sample size.
1. All the figures should be colored. Using different shades of gray doesn't help.
2. The ethical codes of all the institutional ethical approvals are missing.
Comments on the Quality of English LanguageMinor editing is required.
Author Response
COMMENT #1:
“All the figures should be colored. Using different shades of gray…”
RESPONSE #1:
The suggestion is addressed in the relevant figures.
COMMENT #:2.
“The ethical codes of all the institutional ethical approvals…”
RESPONSE #2:
Added accordingly. Please see Lines 348-353.
Round 2
Reviewer 1 Report
Comments and Suggestions for Authors
I recommend accept for publication.
Reviewer 2 Report
Comments and Suggestions for Authors
The manuscript has been improved as requested. No further comments.